# A New Fertilization Approach for Bread Wheat in the Mediterranean Environment: Effects on Yield and Grain Protein Content

Fakir Mathlouthi [1], Roberto Ruggeri [2],*, Angelo Rossini [2] and Francesco Rossini [2]

1   FBSM Nanobiology, Friedrich-Ebert-Anlage 36, 60325 Frankfurt, Germany
2   Department of Agriculture and Forest Sciences, University of Tuscia, Via S. Camillo de Lellis, 01100 Viterbo, Italy
*   Correspondence: r.ruggeri@unitus.it

**Abstract:** Plant biostimulants represent an innovative and sustainable solution to address the challenges of the future agriculture, especially when they are used to improve yield and quality of staple crops. The objective of this research was to study, over three consecutive seasons, the effect of a novel fertilization plan (Thesis 2, T2) on the productivity and protein content of bread wheat (*Triticum aestivum* L.), as compared to the traditional fertilization pattern (Thesis 1, T1), commonly used in Tunisia. T2 was based on the use of a pre-sowing soil bioenhancer (SBE, commercially known as 'Terios') and a topdressing with foliar bio-stimulant (FBS, commercially known as 'Celerios'), obtained by nanotechnology transformation of $Y\text{-}CaCO_3$ minerals (called 'vaterite'); while T1 was based on the use of diammonium phosphate (DAP) at pre-sowing and ammonium nitrate (AN) during the growing season. FBS was applied two times each season and at one rate (3 kg ha$^{-1}$). In each farm and experimental year, the following traits were recorded: plant height (cm), whole aerial biomass (t DM ha$^{-1}$), grain yield (t ha$^{-1}$, 13% moisture content), harvest index, grain weight (mg), spike density (number of spikes per m$^2$), grain protein content (%). T2 protocol slightly, but significantly, increased yield, yield components and grain protein content, while it decreased plant height at harvest. These results suggest that the use of FBS could be of great interest for the cultivation of bread wheat under Mediterranean climatic conditions, as it can make plant nutrients rapidly available even when the uptake from the soil is hindered by water scarcity.

**Keywords:** foliar biostimulant; nanotechnology; sustainable farming; synthetic fertilizers; *Triticum aestivum* L.; vaterite

## 1. Introduction

In North Africa, winter cereals are facing a dramatic reduction of yield potentials because of drought conditions and soil degradation [1,2]. Across the southern shore of the Mediterranean Sea, and specifically in Tunisia, bread wheat (*Triticum aestivum* L.) production is essentially associated with the application of synthetic fertilizers. The traditional fertilization scheme is based on (i) pre-sowing distribution of Diammonium phosphate (DAP) and (ii) nitrogen (N) topdressing using Ammonium nitrate (AN) in late winter—early spring [3]. The cereal sector is considered as the biggest consumer of phosphorous (P) and nitrogen (N)-based fertilizers in Tunisia [4].

For decades, synthetic fertilizers (especially those containing N and P) have been the most important input to boosting the yield of many crops, thus causing serious problems of environmental pollution and health risks [5]. Additionally, fertilization became one of the most expensive inputs in crop management, following the dramatic increases in commodity prices during the last two years. Conversely, cropping systems should nowadays be economically and environmentally sustainable for both farmers and the entire society [6].

These facts, together with the climate change scenario, contributed to raising the risk to global food security [7,8]. In this view, searching for alternative solutions to synthetic fertilizers is urgent as ever.

In a recent publication [9], we demonstrated that bread wheat highly benefited, in terms of early growth, from the pre-sowing application of a biofertilizer, in comparison with the use of commercial DAP. The next step is to verify if also yield and quality traits of bread wheat can be boosted by coupling that pre-sowing fertilization with the foliar biostimulant application.

In the past decades, the use of foliar biostimulants (FBS) in agriculture has markedly increased, as they can trigger several molecular and physiological processes, accompanied by improvement in growth, yield, quality, nitrogen use efficiency and tolerance to biotic and abiotic stresses [10–12]. The general positive effect that FBS can exert on cereal-based cropping systems in the Mediterranean environment can be mainly attributable to the enhancement of metabolic processes such as photosynthesis, modulation of phytohormones, uptake of nutrients and water, and activation of genes responsible for resistance to abiotic stresses and altered plant architecture and phenology [13]. The application of FBS, in particular those obtained using nanotechnology, showed great results in several crops [14–16].

Specifically, the foliar application of micro- and macro-nutrients significantly improved grain yield and quality of diverse cereal crops, including wheat [17–20]. Furthermore, foliar fertilization is highly recommended for rapid assimilation of nutrients, thus leading to an optimization of plant nutrition without jeopardizing environmental sustainability [21].

Calcium is an essential plant nutrient involved in the structure of cell wall and membranes, as well as in mediating the response to numerous abiotic stresses [22,23]. Calcium carbonate ($CaCO_3$) is one of the most common and low-price minerals, widely used in different sectors, including farming. $CaCO_3$ has three anhydrous crystalline polymorphs: 'calcite', 'aragonite', and 'vaterite'. Calcite is the thermodynamically most stable phase, whereas vaterite is the least stable phase, transforming into one of the other two forms. Vaterite particles do not show well-defined morphologies, and usually aggregate into spherical particles [24]. A positive effect of foliar $CaCO_3$ application on physiology, growth, yield, and quality of some crops was recently shown [23,25,26]. However, to the best of our knowledge, very few studies were conducted to assess the effect of foliar $CaCO_3$ application on the agronomic performance and grain quality of the major cereal crops in the Mediterranean environment [27,28].

Many studies have shown that foliar fertilization can reduce the total amounts of fertilizer applied to the soil and reach high fertilizer efficiency, thus limiting both agronomic and environmental issues associated with low nutrient availability, soil acidification and salinization, nutrient leaching and volatilization losses [29]. Moreover, since supplementary foliar application of mineral nutrients can alleviate the adverse effects of abiotic stresses (e.g., heat and drought) on crop yield and quality, this fertilization method will probably be the best way forward in the Mediterranean environment [30,31].

The aim of this study was to assess whether the application of a new fertilization protocol for bread wheat, based on the use of novel and sustainable products would lead to: (i) an agronomic performance (yield and yield-related traits) and (ii) grain protein content that is higher (or at least similar), compared to the traditional use of synthetic fertilizers under typical Mediterranean climatic conditions.

## 2. Materials and Methods

### 2.1. Location and Experimental Design

Field trials were conducted in Medjezel-Bab, governorate of Beja, Northern Tunisia (36°38′58″ N, 9°36′44″ E), under rainfed conditions, in the 2018–2019, 2019–2020, and 2020–2021 seasons (from this point onward referred to as 2019, 2020, and 2021, respectively). Experiments were carried out at the same time in two different farms (hereinafter referred to as F1 and F2). Details on experimental years and cultivation sites are shown in Table 1. Soil

samples were collected in both locations before sowing. The samples, taken at a 0–30 cm depth, were oven dried, grounded, and then analyzed to determine their textural and chemical properties (Table 1).

**Table 1.** Description of experimental seasons and farms (F1 and F2).

| | 2018–2019 | | 2019–2020 | | 2020–2021 | |
|---|---|---|---|---|---|---|
| **Experiment Information** | **F1** | **F2** | **F1** | **F2** | **F1** | **F2** |
| Sowing date | 22 October | 22 October | 21 October | 21 October | 21 October | 21 October |
| Sowing density (seeds m$^{-2}$) | 360 | 360 | 360 | 360 | 360 | 360 |
| Plot size (m$^2$) | 6 | 6 | 6 | 6 | 6 | 6 |
| **Wheater data** | | | | | | |
| Total rainfall (October-May, mm) | 71 | | 82 | | 84 | |
| Mean air temperature (October-May, °C) | 16.1 | | 16.0 | | 15.4 | |
| **Soil features** | | | | | | |
| Clay (Ø < 2 µm, %) | 30 | 40 | 30 | 40 | 30 | 40 |
| Silt (2.0 < Ø < 20 µm, %) | 64 | 58 | 64 | 58 | 64 | 58 |
| Sand (2.0 > Ø > 0.02 mm, %) | 6 | 2 | 6 | 2 | 6 | 2 |
| Soil texture | Silty clay loam | Silty clay | Silty clay loam | Silty clay | Silty clay loam | Silty clay |
| Available P (%) | 0.94 | 0.91 | | | | |
| Total CaCO$_3$ (%) | 3.73 | 2.91 | | | | |
| Organic matter (%) | 2.00 | 1.42 | | | | |
| SiO$_2$ (%) | 30.2 | 42.1 | | | | |
| Al$_2$O$_3$ (%) | 18.7 | 23.2 | | | | |
| Fe$_2$O$_3$ (%) | 9.19 | 11.4 | | | | |
| pH | 7.32 | 7.45 | | | | |

Soil was prepared starting from the end of summer (30 cm depth) and plots were sown in late October using the cultivar "Byrsa". This bread wheat variety, released in 1987, has good resistance to septoria and yellow rust and it is still grown as a high yielding variety in Northern Tunisia [32,33].

A randomized complete block design with three replicates was used. The elementary plots were 5 m long and 1.2 m wide (six rows spaced 0.2 m each other). Two fertilization strategies for each farm and year were compared.

The first fertilization strategy (T1) was the traditional one (which served as control treatment), based on: (i) one pre sowing application of 150 kg ha$^{-1}$ of commercial DAP (NP 18–46), and (ii) 500 kg ha$^{-1}$ of commercial AN (N 33.5), split in two times, the first one at the stage of tillering (250 kg ha$^{-1}$), and the second during the beginning of the stem elongation (250 kg ha$^{-1}$) and when the air temperature starts to increase.

The other fertilization strategy (T2) was based on: (i) one pre-sowing application of 250 kg ha$^{-1}$ of a biofertilizer/soil bio-enhancer (SBE), commercially known as Terios, obtained from a treatment of a natural phosphate ore slurry (see [9] for product details), and (ii) 6 kg ha$^{-1}$ of a FBS based on a natural mineral named "vaterite", split in two times, the first one at the stage of tillering (3 kg ha$^{-1}$), and the second during the beginning of the stem elongation (3 kg ha$^{-1}$) and when the air temperature starts to increase. This latter product, commercially known as "Celerios", was formulated by the company FBSM Nanobiology and had the following composition: Ca$^{2+}$, 29.12%; SO$_4{}^{2-}$, 1.92%; P$_2$O$_5$, 4.79%; NH$_4{}^+$, 4.90%; Fe$^{2+}$, 4.04%; K$_2$O, 4.88%.

The structural and functional effect of this FBS is a result of three consecutive reactions inside the plant leaf:

- Reaction 1:

$$Y\text{-}CaCO_3 \rightarrow CaO + CO_2 + Y \text{ (where Y is a set of microelements composed by Mn, Zn, Fe, Al, Mg)}$$

- Reaction 2:

$$CaO + Y \rightarrow Ca^{2+} + O^{2-} \text{ (Photosynthesis)} + Y$$

- Reaction 3:

$$CO_2 \rightarrow C^{4+} \text{ (carbon, biomass)} + 2\, O^{2-} \text{ (Photosynthesis)}.$$

While the release of $O_2$ in reaction 2 and 3 and $CO_2$ in reaction 1, will boost the photosynthesis, the other elements (e.g., $Ca^{2+}$) will lead to plasmodesma regulation of the treated plant [34]. Finally, C, in reaction 3, will help with growing plant biomass.

### 2.2. Measurements of Agronomic Traits and Grain Protein Content

In each farm and experimental year, the following traits were recorded: plant height (cm), whole aerial biomass (t DM ha$^{-1}$), grain yield (t ha$^{-1}$, 13% moisture content), harvest index, grain weight (mg), spike density (number of spikes per m$^2$), grain protein content (%).

A ruler from the soil surface to the tip of the spike, excluding the awns, was used to measure plant height.

The whole aerial biomass and harvest index (HI) were determined at maturity, sampling an area of 1 m$^2$ and following the indications by [35].

The determination of grain yield was accomplished by threshing the entire plot (with the exclusion of external rows) and weighing the harvested grains. Subsequently, a subsample of grains (60 g) was oven dried (70 °C, until constant weight) to determine the grain moisture content.

Grain weight was obtained from weighing two 200-seed samples plot$^{-1}$.

Spike density was determined counting the number of spike bearing culms from the same sample used to determine the aerial biomass.

Grain samples (250 g) from each plot were ground through a PerkinElmer LM-3610 grinder (PerkinElmer Health Sciences Canada Inc., Winnipeg, MB, Canada) into a 1.0 mm particle size. Grain protein content was determined scanning the samples with a benchtop NIR (Near Infrared Reflectance) spectrometer DS2500 flour analyzer (FOSS-DS2500; FOSS Electric A/S, Hillerød, Denmark).

### 2.3. Statistical Analysis

A three-way analysis of variance (ANOVA) was performed using R (version 3.5.2) to test the main effects of the year, farm, fertilizer, and their interactions [36]. Treatment means were separated using Fisher's protected least significant differences test at a probability level of 0.05.

## 3. Results

### 3.1. Plant Height and Aerial Biomass

Plant height was significantly affected by the cultivation year and fertilization method, while aerial biomass was affected only by the cultivation year (Table 2). As shown in Figure 1A, the T1 fertilization strategy significantly increased the plant height. Specifically, plots treated with traditional fertilization strategy (T1) showed wheat plants 7 cm taller than plants treated with the new approach (T2). Difference in cultivation site did not significantly affected the height of wheat, while the climatic conditions of the 2021 has significantly increased the plant height as compared to 2019 and 2020 (2.3 cm and 3.2 cm taller than 2020 and 2019, respectively).

**Table 2.** Results from the analysis of variance for the recorded traits.

| | Plant Height | Aerial Biomass | Yield | Spike Density | Grain Weight | Harvest Index | Grain Protein Content |
|---|---|---|---|---|---|---|---|
| Year | *** | *** | *** | *** | *** | ** | *** |
| Farm | ns | ns | * | ns | * | ns | * |
| Fertilization | *** | ns | ** | ** | * | ns | * |

Levels of significance: *** < 0.001; ** < 0.01; * < 0.05; ns: not significant. Interactions are not reported since they were not significant.

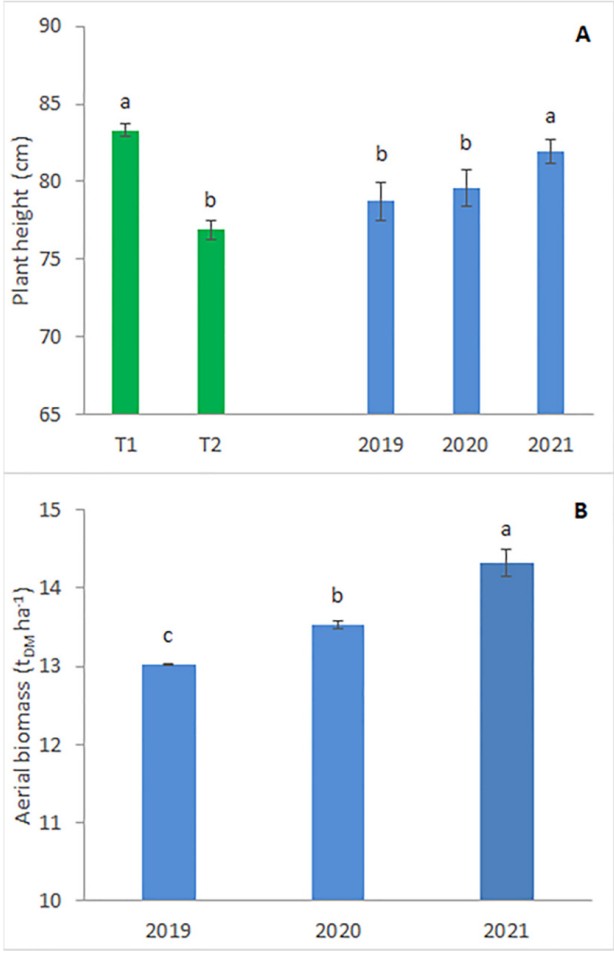

**Figure 1.** Plant height (**A**), and aerial biomass (**B**) for each fertilization method (T1 and T2), and growing season. Error bars represent the standard error of the means; letters above the histograms correspond to the ranking of the Fisher's protected test at $p < 0.05$.

Consistent with plant height, the whole aerial biomass significantly increased moving from 2019 to 2021 (Figure 1B). In more detail, aerial biomass increased by 4% from 2019 to 2020 and by 6% from 2020 to 2021.

### 3.2. Yield, Yield Components and Harvest Index

Yield and grain weight were significantly affected by all the tested treatments (year, location, and fertilization method) and spike density by year and fertilization method, while harvest index was affected only by the cultivation year (Table 2). As shown in Figure 2A, the T2 treatment slightly (but significantly) enhanced grain yield, as compared to T1 treatment. Specifically, the adoption of the new fertilization method resulted in a 0.7% yield increase, as compared to control treatment (5.55 t ha$^{-1}$ vs. 5.59 t ha$^{-1}$). The effect of T2 treatment on grain yield reflects the response of yield components to the same treatment

(Figure 2B, C). The new fertilization approach enhanced the spike density by about 0.9% (586 vs. 581 spikes $m^{-2}$) and the mean grain weight by 1.7% (46.9 mg vs. 46.1 mg), as compared to the use of traditional synthetic fertilizers.

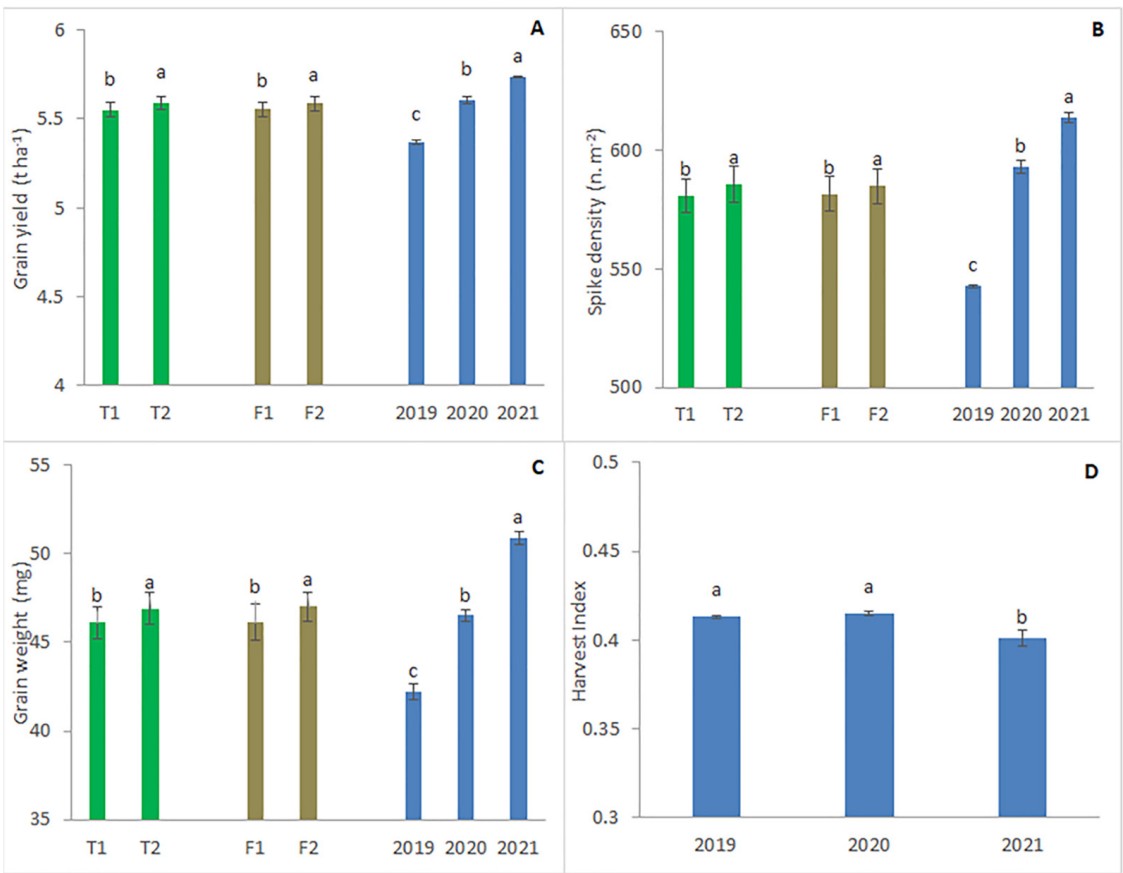

**Figure 2.** Grain yield (**A**), spike density (**B**), grain weight (**C**), and harvest index (**D**) for each fertilization method (T1 and T2), cultivation site (F1 and F2) and growing season. Error bars represent the standard error of the means; letters above the histograms correspond to the ranking of the Fisher's protected test at $p < 0.05$.

As clearly appears in Figure 2A–C, grain yield and its components were markedly affected by the different climate conditions across the three growing seasons and, to a lesser extent, by the cultivation site. Particularly, the most productive growing season was 2021, with a 7% and 2% yield increase as compared to 2019 and 2020, respectively. The difference is enlarged when considering the yield-related traits. Spike density in 2021 was 13% and 3.5% higher than in 2019 and 2020, respectively. Mean grain weight in 2021 was 21% and 9.5% higher than in 2019 and 2020, respectively.

Difference in cultivation site produced small yield and yield components gap (less than 1% for grain yield and spike density; 2% for mean grain weight).

Finally, harvest index showed similar results in 2019 and 2020 (0.413 vs. 0.415, respectively), while it slightly decreased in 2021 (0.401, Figure 2D).

### 3.3. Grain Protein Content

Albeit grain protein content was significantly affected by all the tested treatments (year, location and fertilization method, Table 2), it markedly varied just across the growing seasons (Figure 3). In more detail, the new fertilization pattern (T2) and F2 slightly increased the grain protein content of bread wheat (1.7% and 0.8%, respectively), as compared to the traditional application of synthetic fertilizers (T1) and F1. As regards growing season, grain

protein content was found to be below 12% in 2019 (11.7%) while it increased by 5% in 2020 (12.3%) and by 7% in 2021 (12.5%).

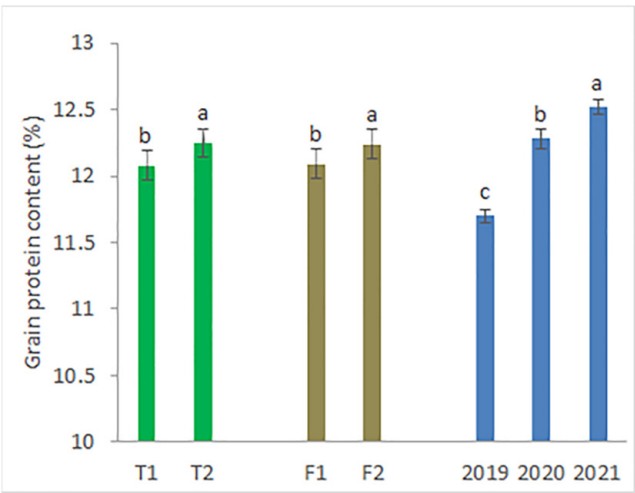

**Figure 3.** Grain protein content for each fertilization method (T1 and T2), cultivation site (F1 and F2) and growing season. Error bars represent the standard error of the means; letters above the histograms correspond to the ranking of the Fisher's protected test at $p < 0.05$.

## 4. Discussion

Overcoming the excessive use of chemical inputs is a major target for the agriculture industry over the coming years [37,38].

As shown in this study, the application of a novel and sustainable approach for the fertilization of bread wheat can be a valid alternative to the traditional fertilization scheme based on the use of synthetic fertilizers or, at least, can help to reduce their use in drought-prone environments.

As expected, the comparison between two fertilization methods, applied at the same growth stages, led to relatively small differences in terms of yield, yield components and grain protein content, even if these differences were always significant. The positive effect of N and P mineral fertilization on wheat yield and grain quality under Mediterranean climatic conditions is well supported in the scientific literature [19,39,40]. However, our objective was not to obtain remarkable differences between the two fertilization protocols, but rather to verify that the new fertilization strategy did not negatively affect grain yield and protein content, as compared to the traditional application of synthetic fertilizers.

In this study, we found the traditional fertilization method (T1) significantly increased plant height as compared to T2, while no difference was detected between T1 and T2 when considering the aerial biomass. This can be explained by a higher tillering ability of plants treated with the T2 method, as compared to that of plants treated with the T1 strategy. The emission of new tillers compensated the reduction in aerial biomass due to a lower plant height. This was mainly due to the effect of pre-sowing fertilization which originated, at 70 days after emergence, the following positive traits, as compared to the traditional use of DAP: a significantly higher number of tillers, a greater leaf area index, an improved leaf P concentration [9]. All these traits are known to positively affect the yield and yield components of wheat [41–43]. A lower plant height can be considered, in any case, a positive trait, because it considerably reduces the risk of lodging [44].

Since grain yield is not only determined by the carbohydrates stored in the early growth stages, but also by the photosynthetic products after heading, spring topdressing represents a crucial strategy to enhance both wheat production and quality [19]. The importance of this practice is underlined also by the fact that, while wheat plants have the ability to adapt to early stress, yields are severely depressed when stressors occur from boot stage onwards [45].

In our study, the application of a FBS resulted in a better yield performance of wheat plants and higher grain protein content, as compared to soil AN distribution.

Many other studies showed that foliar fertilizers were useful for the improvement of nutrient utilization efficiency, yield, and quality of different crops [29].

Specifically, other authors found that foliar application of $Ca^{2+}$ and microelements markedly enhanced the yield and yield-related traits of wheat [46–48]. This could be explained by the beneficial effect of calcium on the plant physiological and metabolic processes that lead to an increased photosynthesis rate, chlorophyll content and antioxidant activity [27]. Moreover, calcium is known to act as a regulator of many physiological and biochemical processes in response to abiotic stresses in plants [49,50]. This main characteristic can play a crucial role in the current climate change scenario, especially in the Mediterranean region. In recent years, observing wheat suffering drought and heat stress during its growing cycle is becoming more and more frequent [51,52]. A high dose of nitrogen, given to a water stressed crop, could lead to a depression of grain yield and quality. This phenomenon is called "haying off", and it is caused by a depressed capacity of the plant to absorb water from the soil during the post anthesis period and to an inadequate re-translocation of the reserves [53,54]. Therefore, topdressing of wheat in the Mediterranean environment should be rethought accordingly.

The distribution of a great amount of nitrogen to soil when the plants are not able to use it, makes no sense economically and poses many environmental threats [55]. Conversely, the use of foliar fertilization as an efficient supplement to the usual crop management practices, can help Mediterranean farmers to achieve environment-friendly crop production without renouncing the socioeconomic benefits [30,31].

Beside yield, grain quality is often the other feature in determining the income of wheat growers. The grain protein content is one of the main quality traits of bread wheat, influencing both its nutritional value and end use [56]. Among other factors, grain protein content is known to be influenced by nitrogen rate and form, time of nitrogen application, and residual soil nitrogen [19,57]. In this study, we demonstrated that two applications of a FBS can lead to a grain protein content higher than that obtained using commercial AN fertilizer.

The positive effect of foliar fertilizers containing a mixture of macro- and micro-elements (and other substances) on grain protein content has been reported by other studies conducted in the Mediterranean environment and attributed to a complex interactions between elements and to the activation of beneficial biochemical processes [58,59]. Although complex mixtures of plant biostimulants were found to be more effective than single elements, a deep understanding of their mechanism of action remains an interesting challenge for the future.

In this study, we found a marked effect of the growing season on the collected data. The variation over the years can be principally ascribed to the different levels of precipitation that occurred during the growing cycle. In 2019, total rainfall during the growing cycle of wheat (from October to May) was 13% and 15% lower than in 2020 and 2021, respectively. In more detail, soil drought during the early growth stages has strongly limited the agronomic performance of wheat in 2019 [9]. In dry regions, the amount of precipitation during the vegetative phase of wheat was the determining factor in achieving a satisfying yield response [60].

## 5. Conclusions

The application of an alternative fertilization protocol based on the use of sustainable products (soil bio enhancer and foliar biostimulant), clearly enhanced grain yield and protein content of bread wheat in North Africa.

This study proved that the use of foliar biostimulants can help farmers to reduce the amount of chemical inputs, especially synthetic nitrogen fertilizers, without reducing the yield and grain protein content under Mediterranean climatic conditions.

Additional studies are required to understand better the mechanism of action of this foliar biostimulant, and to verify whether this alternative fertilization program will be suitable for other crops.

**Author Contributions:** Conceptualization, F.M. and F.R.; methodology, F.M.; resources, F.M.; data curation, R.R. and F.R.; writing—original draft preparation, R.R., A.R., F.M., F.R.; writing—review and editing, R.R.; supervision, F.M. and F.R. All authors have read and agreed to the published version of the manuscript.

**Funding:** This research received no external funding.

**Data Availability Statement:** The data presented in this study are available on request from the corresponding author.

**Conflicts of Interest:** The authors declare no conflict of interest.

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
