# Peer review of "A New Fertilization Approach for Bread Wheat in the Mediterranean Environment: Effects on Yield and Grain Protein Content"

_agronomy, doi:10.3390/agronomy12092152_

Round 1

Reviewer 1 Report

The current research required addation experiments 

Reviewer 2 Report

This is a very interesting study on “A new approach to bread wheat fertilization in the Mediterranean environment,” and the authors have collected unique datasets. The paper is generally well written. However, in my opinion, the paper has some shortcomings in regard to available data and text, and I feel this unique dataset has not been utilized to its full extent. Below I have provided numerous remarks on the text as it is often vague and long-winded. In several instances, I also suggested citing more relevant and recent literature. Furthermore, I have made line by line additional suggestions for more in-depth analyses of the data. Key critical points are:-

Line by line comments to authors for the revisions of the current study

Title

Please conclude the key findings or the main contents of the study and make a meaningful title; in its current form, it will not attract readers of the same field. E.g., Enhanced production of yield and protein contents in bread wheat using a new approach of fertilization in the Mediterranean environment.

Abstract

Line 14, plz name biostimulant at its first-time used in each section

Line 21-23, Please mention a logical reason why FBS could be of great interest in a Mediterranean environment.

Line 26, 27= Remove those keywords which are used in the title

Introduction

Line 30-33. Rephrase the sentence and add a reference.

Line 38= Add a reference to strengthen your thought.

Line 45= A reference must be added at the end of the sentence to strengthen the mentioned lines in the relevant data,

Line 55= What physiological processes are triggered by FBS in the cropping system in question

Line 74, plz add references and strengthen your data with the latest references.

Line 74= A paragraph must be added at this place to materialize the importance of these fertilizations with reference to soil and climate in the Mediterranean.

Line 75-78= This is confusing; please clear your objectives one by one in writing so that the readers can easily understand the essence of the study.

Materials and methods

Line 91= This pattern of adding reference is not up to the mark, and as per journal policy, please mention here the reference in parentheses and in number but keeping in view the previous numbers.

Line 89 to 91= Rephrase this sentence.

Line 128= Does it need to mention the formulas for the assessment of these parameters?

In complete materials and methods, you have mentioned the protein contents determination, neither procedure nor the formula. How have you assessed these contents?

If you have forgotten, then plz mention add up the relevant information.

Results

Line 151. The start of the results section is not up to the mark; when submitting an article in such a good journal, then please make sure to keep your writing up to mark. Rather I would suggest taking help from a native English speaker to help make better the startup of the line at various places.

As per your results, these experiments have not shown promising results for protein contents. Can you explain this?

Discussion:-  In general I have seen that you have only mentioned 7 to 9 support references yet I hope on revisions authors will work more in-depth to improve the discussion section as per the policy. In my suggestion, authors must provide a 2 to 3 pages discussion of all the experiments with logic and exclusively discuss in detail the benefits of this pattern of fertilization with respect to the environment by applying this model of study. I will also suggest improving the English quality of this discussion section, and again I will repeat to discuss the protein contents ratio and discuss in detail why significant improvement has not been seen in the experiments rather, it is a good model to deal with wheat issues.

Improve the English write-up.

Keeping all in view, I will suggest major revisions and recommend that these are very necessary revisions to keep this article in good form.

Reviewer 3 Report

Dear Authors,

Your research paper is interesting because it shows a new approach to bread wheat fertilization in the Mediterranean environment. The topic you raise is extremely important and topical. This paper is prepared in the usual way for scientific work. It is prepared very carefully. The language appears to be correct, but I don't feel qualified to judge about the English language and style. I have no comments and recommend it to be printed in this form.

Good luck!

Sincerely yours

Reviewer

Round 2

Reviewer 1 Report

Dear Editor, 

I think the authors do all the recommended comments and its fine for accept 

Reviewer 2 Report

Dear Authors

I believe that manuscript is now significantly improved. It can be accepted in its current form.